# Compliance with Antibiotic Prophylaxis in Obstetric and Gynecological Surgeries in Two Peruvian Hospitals

**DOI:** 10.3390/antibiotics12050808

**Published:** 2023-04-25

**Authors:** Kovy Arteaga-Livias, Vicky Panduro-Correa, Jorge L. Maguiña, Jorge Osada, Ali A. Rabaan, Kiara Lijarza-Ushinahua, Joshuan J. Barboza, Walter Gomez-Gonzales, Alfonso J. Rodriguez-Morales

**Affiliations:** 1Maestría en Epidemiologia Clinica y Bioestadistica, Universidad Científica del Sur, Lima 15014, Peru; 2Facultad de Medicina, Universidad Nacional Hermilio Valdizán, Huánuco 10000, Peru; 3Escuela de Medicina, Universidad San Juan Bautista, Lima 15067, Peru; 4Molecular Diagnostic Laboratory, Johns Hopkins Aramco Healthcare, Dhahran 31311, Saudi Arabia; 5College of Medicine, Alfaisal University, Riyadh 11533, Saudi Arabia; 6Department of Public Health and Nutrition, The University of Haripur, Haripur 22610, Pakistan; 7Sociedad Científica de Estudiantes de Medicina (SOCIEM-HCO), Universidad Nacional Hermilio Vadlizán, Huánuco 10000, Peru; 8Vicerrectorado de Investigación, Universidad Norbert Wiener, Lima 15046, Peru; 9Escuela de Medicina-Filial Ica, Universidad Privada San Juan Bautista, Ica 11001, Peru; 10Grupo de Investigación Biomedicina, Faculty of Medicine, Fundación Universitaria Autónoma de las Américas, Pereira 660005, Colombia; 11GIlbert and Rose-Marie Chagoury School of Medicine, Lebanese American University, Beirut P.O. Box 36, Lebanon

**Keywords:** compliance, antibiotics, surgical site infections, prophylaxis, gynecologic surgeries

## Abstract

Introduction: Surgical site infections (SSI) can be as high in gynecology and obstetrics surgeries compared to other areas. Antimicrobial prophylaxis is an effective tool in the prevention of SSIs; however, it is often not adequately administered, so this study aimed to understand the compliance and factors associated with the use of the clinical practice guidelines for antibiotic prophylaxis in gynecological surgeries in two hospitals in the city of Huanuco, Peru. Methods: An analytical cross-sectional study of all gynecologic surgeries performed during 2019 was performed. Compliance was determined based on the antibiotic chosen, dose, administration time, redosing, and prophylaxis duration. Age, hospital of origin, presence of comorbidities, surgery performed, as well as its duration, types of surgery, and anesthesia were considered as related factors. Results: We collected 529 medical records of patients undergoing gynecological surgery with a median age of 33 years. The prophylactic antibiotic was correctly indicated in 55.5% of cases, and the dose was correct in 31.2%. Total compliance with the five variables evaluated was only 3.9%. Cefazolin was the most commonly used antibiotic. Conclusion: Low compliance with the institutional clinical practice guidelines for antibiotic prophylaxis was identified, showing that antimicrobial prophylaxis in the hospitals studied was inadequate.

## 1. Introduction

Surgical site infections (SSIs) are the most frequent nosocomial infections in developing countries [1,2]. SSIs pose a significant burden in terms of morbidity, mortality, and hospital costs. Patients who develop an SSI have a twofold increased risk of death and up to a 6-fold increased chance of readmission after discharge, as well as a twofold increased risk of admission to intensive care, with a concomitant risk of prolonged hospital stay [3].

SSIs are a major source of healthcare expenditure in the Region of the Americas, with estimates suggesting that they cost between USD 3.5 and 10 billion annually. In addition to the direct costs of treatment, SSIs also result in approximately 90,000 readmissions each year, incurring an additional cost of USD 700 million [4].

It has been described that cesarean delivery is associated with a 20 times higher risk of infection compared to vaginal delivery, with a rate of infection reported to be 1–25% [5] in developed countries and an estimated 10–48% of women delivering via cesarean in Sub-Saharan Africa [6]. These surgeries are complicated by SSIs, endometritis, and urinary tract infections if prophylactic antibiotics are not used [7]. Of the total number of gynecological surgeries, SSIs may develop in up to 10% of cases [8]. The incidence of SSI can be as high in procedures performed in gynecology and obstetrics when compared to other areas [1,9,10].

Antibiotic prophylaxis is defined as a brief course of antibiotic use initiated close to the surgical procedure and aimed at reducing the likelihood of developing SSIs [11]. Antibiotic prophylaxis is aimed at obtaining bactericidal levels in the tissue at the time of the incision and at reducing the bacterial load intraoperatively [12]. Although antibiotic prophylaxis has proven to be an effective tool in preventing SSIs, in Peru and many other countries, its use is not regulated [13]. In hospitals with procedural guidelines, medical prescribers often do not follow them adequately. 

Compliance with clinical practice guidelines is crucial for ensuring the appropriate and effective use of antibiotics. These guidelines offer recommendations for the selection, timing, and duration of antibiotic prophylaxis based on patient characteristics and the surgical procedure [14]. However, reports have shown that broader spectrum antibiotics, unnecessary combinations of antibiotics, suboptimal timing, and prolonged duration of surgical antibiotic prophylaxis are being used [3].

Mistakes in the administration of antibiotic prophylaxis are common, with errors such as delayed initiation of antibiotics after the incision or even after the entire surgical procedure, use of antibiotics not recommended for a specific surgery, and inadequate dosing being reported. Patients admitted to gynecology and obstetrics wards are at higher risk of not receiving appropriate antibiotic prophylaxis for more than 24 h [11].

The inappropriate use of antibiotic prophylaxis and the overuse of antibiotics have resulted in numerous negative outcomes, including increased adverse reactions, hospital cost overruns, bacterial resistance, and superinfections [12]. The World Health Organization (WHO) has identified antibiotic resistance as one of the top ten global public health threats facing humanity today [15]. Non-compliance with the appropriate use of antibiotics can be caused by various factors, including lack of awareness, knowledge gaps, misperceptions, cultural or institutional barriers, and competing priorities. For these reasons, it is essential to promote the rational and efficient use of antimicrobials to ensure their appropriate use [16].

Enhancing compliance with clinical practice guidelines for antibiotic prophylaxis is a crucial strategy for reducing the inappropriate use of antibiotics. To address incorrect use, it is essential to judiciously follow the clinical practice guidelines that have already been published by various societies and institutions. By promoting adherence to established guidelines, healthcare providers can ensure that antibiotics are used appropriately and in the best interests of patient health [17,18].

The purpose of this study was to assess compliance with the American Society for Hospital Pharmacists (ASHP) clinical practice guidelines for antibiotic prophylaxis published in 2010 in two hospitals located in Huanuco, Peru. Additionally, we aimed to identify factors associated with compliance in these settings.

## 2. Results

During the study period, 529 medical records were collected from patients who underwent gynecological surgery with a median age of 33 years, the cesarean section being the most frequently performed surgery in 57.1% of the cases. The prophylactic antibiotic was correctly indicated in 55.5% of cases, and the dose was correct in 31.2%. Complete compliance with the five variables evaluated was only 3.9% (Table 1).

With regards to the prophylaxis administered, our findings reveal that 94.5% of patients were prescribed antibiotics, with 38.2% receiving antibiotic prophylaxis. Cefazolin was the most frequently used antibiotic in both cases (Table 2).

Finally, when comparing patient characteristics with the type of compliance, we observed that the surgery performed, the type of procedure, and the type of anesthesia were related to total compliance, while the hospital, the surgery performed, and the type of procedure to acceptable compliance (Table 3).

## 3. Discussion

The use of antimicrobial prophylaxis in obstetric and gynecologic surgeries varies considerably according to geographical areas and socioeconomic and cultural contexts. The present study found that, of the total number of surgeries, antibiotics were used in 94.5%; however, the indication for antibiotic prophylaxis only corresponded to 38.2%. This is different from what was found by Uppendahl and Gil, who, in their respective studies, found a frequency of use of prophylaxis above 95% [19,20], even lower than that described by Romero, in a Latin American context, where the overall use of antimicrobial prophylaxis was 69.9% [13]. The low adherence to prophylaxis in Peru is notorious. In a Latin American study of point prevalence of antibiotic use, Peru together with Mexico were the countries with the lowest adherence [21]. 

It has been shown that the publication and follow-up of clinical practice guidelines improve the proper prescription of antimicrobial prophylaxis in surgery in hospitals where they have been implemented [22,23,24]. A study evaluating surgeons’ surgical prophylaxis practices found that gynecologists administered antimicrobial prophylaxis at the correct time only 51.5% of the time, and up to 35.5% administered it together with the incision [25]. At the same time, anesthesiologists appeared to adhere more to antimicrobial prophylaxis in cesarean-type surgeries [26]. Under these considerations, most hospitals now have an anesthesiologist administering antimicrobial prophylaxis.

While in surgeries that used antibiotics in general, cefazolin corresponded to only 37.6% of all indications; in cases where antimicrobial prophylaxis was indicated, cefazolin was used in 57.4%. A study in Jordan found that cefazolin was 85.4%, but in its correct dose, it was only 62.3% [27], while in Latin America, the use of this antibiotic was also relatively high [17]. The rest of the antibiotics used vary according to regions and studies: ceftriaxone, metronidazole, azithromycin, aminoglycosides, etc., making their comparison difficult and unnecessary.

The present study has shown that the use of multiple antibiotics was more frequent in cases of treatment compared to prophylaxis. Specifically, 17% of cases involved the use of more than two antibiotics in treatment, while only 2.5% did so in prophylaxis. This finding highlights the importance of rational and conscientious antibiotic prophylaxis. Physicians who correctly prescribe antibiotic prophylaxis tend to use only one antibiotic, which is a significant advantage. However, previous studies have shown that antibiotic combination is still prevalent in some cases of prophylaxis, with Alemkere reporting that 39% of prophylaxis cases used antibiotic combinations [11]. In another study, Abubakar found redundant combinations in 99% of cases, which decreased to 76% after educative intervention [3]. The high proportion of antibiotic combinations (such as Clindamycin + Gentamicin or Ceftriaxone + Clindamycin) may stem from a fear of multiple bacterial etiologies causing SSIs. However, some combinations (such as Ceftriaxone + Clindamycin + Gentamicin, Ceftriaxone + Gentamicin, Cefazolin + Gentamicin, or Cefazolin + Gentamicin + Chloramphenicol) have similar coverage without an adequate rationale for their use.

Even after discharge, a significant proportion of patients were found to continue using antibiotics. Alexiou’s research showed that 15% of gynecologists prescribed antibiotic prophylaxis for two days or more [25], while Aulakh and Romero’s studies reported that 100% of patients received antibiotics after surgery [13,28]. The overuse of antibiotics is a global concern, and efforts to improve prescribing practices should focus on reducing this unnecessarily prolonged use. These practices not only contribute to the growing problem of antibiotic resistance but also increase the risk of adverse effects and unnecessary healthcare costs.

The study revealed that the highest frequency of compliance was observed in the choice of the right antibiotic, while the lowest frequencies of compliance were observed in relation to redosing and duration of prophylaxis. A study in Israel showed that 95.6% of patients received the correct antibiotic [29]. In contrast, a study in Spain identified the timing of initiation and correct antibiotic selection as the main causes of non-compliance [20]. Importantly, differences in compliance rates could be attributed to the presence of antibiotic optimization and monitoring units in some hospitals. These units are not present in either of the two hospitals studied.

The study’s findings regarding compliance with prophylaxis protocols in surgical patients are concerning. Total compliance with all five correct parameters was found to be only 3.9%, despite the fact that only clean and clean–contaminated surgeries were included in the analysis—all of which should have received prophylaxis instead of antibiotic treatment. When considering the appropriate choice of antibiotic and dose, as well as an additional correct parameter, acceptable compliance rates only reached 9.8%. These results are consistent with those of previous studies. Abdel et al. reported overall adherence of only 2.7% in all surgeries [27]. In a large cohort study of over 400,000 cases in the United States, antibiotic adherence was found to be only 35.9% [30].

The study found that the type of surgery was a statistically significant factor associated with both total and acceptable compliance, with cesarean sections showing better adherence to prophylaxis protocols compared to other types of surgeries. These findings are consistent with those of previous studies. For example, Uppendahl reported that myomectomy, laparoscopy, and ectopic pregnancy procedures received agents that were contrary to recommendations in up to 96% of cases [19]. In contrast, Tietel found that adherence to prophylaxis protocols was over 90% in cesarean sections [29]. Kremer, on the other hand, found that adequate adherence to antimicrobial prophylaxis decreased significantly [8]. These results highlight the importance of considering the type of surgery when designing interventions to improve compliance with prophylaxis protocols. Cesarean sections, in particular, may require less specific attention due to their higher compliance rates compared to other types of surgeries.

The results indicate that there is a higher level of adherence to prophylaxis guidelines in emergency surgeries compared to scheduled surgeries. This is supported by Abubakar’s study, which demonstrates that the correct administration of prophylaxis is more frequent in emergency surgeries [1]. However, this finding is not consistent across all studies. For example, research conducted in Jordan has shown that emergency surgeries carry a greater risk of prolonged antibiotic therapy [27], while inadequate use of prophylaxis is more prevalent in emergency surgeries in the United States [30]. Despite some variability in the use of prophylaxis guidelines in emergency surgeries across different regions, the evidence suggests that compliance is generally higher in emergency surgeries than in scheduled surgeries.

There were statistical differences regarding the type of anesthesia, being better in general anesthetics. This differs from that shown by Polla, where local anesthesia was more likely to receive adequate antibiotic prophylaxis [31]. These differing results suggest the need for further research to better understand the relationship between the type of anesthesia and antibiotic prophylaxis in order to optimize patient outcomes.

The hospital of origin was found to have statistical differences only for acceptable compliance but not for total compliance. Previous studies, such as Brubaker’s, have found geographic differences in the likelihood of receiving the correct antibiotics, while Hamdaoui found that the proportion of antibiotic prophylaxis increased with the number of procedures, particularly in the case of abortion prophylaxis [7,32]. These results suggest that even factors as little considered as hospital and geographic location may influence compliance with antibiotic prophylaxis guidelines.

The present study has certain limitations that must be pointed out. First, the statistical results may be affected due to the low proportion of cases with adequate compliance. However, this was partly mitigated by categorizing compliance as complete or acceptable, resulting in consistent results among factors associated with compliance. Second, limited data on certain groups of related factors may restrict the generalizability of these findings to the broader population. Moreover, the study did not consider variations in surgical techniques or practices among surgeons, which could also affect compliance rates, nor the presence of pre-eclampsia or HELLP syndrome, which, although they do not usually require antibiotics beyond prophylaxis, patients with these conditions tend to have longer hospital stays, increasing the risk of iatrogenic infections and the need for antibiotic treatment. Finally, the study was conducted in only two institutions, which could limit the applicability of the findings to other healthcare settings or regions.

## 4. Materials and Methods

An analytical cross-sectional study with retrospective data was conducted in the gynecology and obstetrics services of all surgeries that received antimicrobial prophylaxis. Antimicrobial prophylaxis was defined as the use of antibiotics before surgery to prevent infections associated with the surgical site. Age, hospital of origin, presence of comorbidities, type of surgery performed, duration of surgery, types of surgery, and anesthesia were all considered as related factors for analysis. Elective and emergency procedures were included.

Data collection was performed in the gynecology and obstetrics services of the two leading hospitals of the city of Huánuco “Hospital Regional Hermilio Valdizan” and “Hospital II EsSalud-Huanuco” from 1 January to 31 December 2019. The selected hospitals serve as a point of reference for the entire region of Huanuco, offering gynecology and obstetrics services to a population of 375,000 women with over 80 available beds.

The data collection process was carried out over a period of four months, from January to April 2022, during which all medical records and operative reports from both hospitals were reviewed. However, to ensure the validity of the study, certain patients were excluded from the analysis. Specifically, those who had previously used antibiotics during hospitalization or undergone surgery within the preceding three months were excluded, as these factors may have influenced the use of antimicrobials not recommended for antibiotic prophylaxis. In addition, procedures performed on individuals with severe immunosuppression, those who had been hospitalized for more than 48 h at the time of the procedure, individuals with suspected or confirmed colonization by resistant microorganisms, and those surgical histories and reports that did not present complete data related to prophylaxis were also excluded from the analysis.

The investigators followed the U.S. Center for Disease Control (CDC) wound classification to include only clean and clean–contaminated wounds: Class I/Clean wounds are those which are uninfected and do not encounter any inflammation. Additionally, these wounds do not enter the respiratory, alimentary, genital, or uninfected urinary tract. Clean wounds are typically closed and, if needed, drained with closed drainage. If operative incisional wounds are a result of non-penetration (blunt) trauma and meet the criteria, they should be included in this category. Class II/Clean–contaminated wounds refer to operative wounds where the respiratory, alimentary, genital, or urinary tracts are entered under controlled conditions, and there is no unusual contamination. This category includes operations that involve the biliary tract appendix, vagina, and oropharynx, provided there is no evidence of infection or a significant breach in technique.

All parameters were evaluated individually, according to the Guide for Surgical Prophylaxis published by the Institute for the Evaluation of Health Technologies and Research (IETSI) of the Peruvian Social Security (EsSalud). The IETSI guidelines follow the antibiotic prophylaxis guidelines of the United Kingdom and the American Society for Hospital Pharmacists (ASHP).

The study established the following parameters:Antibiotic type: this was deemed correct if the antibiotics used were recommended by prophylaxis guidelines and incorrect if any other antibiotic was administered;Appropriate dose: the dose was classified as correct if it corresponded to the dosage recommended by the prophylaxis guidelines and incorrect if any other dosage was indicated;Time of administration: this parameter was considered correct if the antibiotic was administered within 60 min prior to the incision and incorrect if it was administered beyond 60 min or after the incision;Duration of prophylaxis: the prophylaxis duration was deemed correct if the antibiotic was stopped on the same day of the surgery and incorrect if the antibiotic continued to be administered in the following days;Need for redosing: considered as correct to the administration of an additional dose of antibiotic during a surgical procedure when the procedure is prolonged beyond the recommended duration for the chosen antibiotic and was considered incorrect if the antibiotic was not re-administered or was administered before the recommended time.

For the purposes of the study, acceptable compliance was defined as prescriptions that adhered to the correct choice of antibiotic, appropriate dosage and redosing, and adherence to the recommended time or duration of prophylaxis. Meanwhile, total compliance was achieved when all five requirements were met for every surgery.

Stata v16 (StataCorp) was used for data analysis. The descriptive statistics used were mean, median, standard deviation, and percentiles according to their distribution for continuous variables and percentages for categorical variables. Only histories with complete data were used. Bivariate analysis was performed to explore the possible variables associated with total and acceptable compliance. The chi-square test was used to compare categorical variables, and the Mann–Whitney U test was used for nonparametric quantitative variables. The statistical significance level of *p* < 0.05 was predetermined. 

This research was conducted in accordance with the ethical principles emanating from the regulatory standards for human health research and the Declaration of Helsinki of the World Medical Association and all its amendments. The Universidad Cientifica del Sur, through Research Ethics Committee, granted approval for the project, identified as number 026-2019. As the study relied on medical records, obtaining informed consent was not deemed necessary. This article represents the thesis submitted for a master’s degree in clinical epidemiology and biostatistics at the Universidad Científica del Sur.

## 5. Conclusions

We conclude that antibiotic prophylaxis in gynecological surgeries in two hospitals in Huánuco is inadequate, and procedures have low compliance with institutional clinical practice guidelines. Cesarean sections and emergency surgeries are statistically related to better antibiotic prophylaxis prescriptions.

## Figures and Tables

**Table 1 antibiotics-12-00808-t001:** Clinical and prophylaxis characteristics of patients undergoing gynecological surgeries (*n* = 529).

Characteristics	Frequency	Percentage
Age		
Median (IQR)	33 (28–38)	
Hospital		
EsSalud	308	58.2
Minsa	221	41.8
Comorbidities		
HIV	1	0.2
Anemia	1	0.2
Diabetes	2	0.3
Hypertension	4	0.8
No	521	98.5
Surgery		
Cesarean	302	57.1
Uterine curettage	129	24.4
Hysterectomy	55	10.4
Others	43	8.1
Type of procedure		
Emergency	319	60.3
Programmed	210	39.7
Anesthesia type		
Regional	357	67.5
General	154	29.1
Sedoanalgesia	18	3.4
Length of surgery (min)		
Median (IQR)	90 (60–130)	
Correct prophylactic antibiotic		
Yes	112	55.5
No	90	44.5
Correct prophylactic dose		
Yes	63	31.2
No	139	68.8
Time of proper administration		
Yes	105	51.9
No	97	48.1
Redosing		
Yes	1	6.3
No	15	93.7
Correct length of prophylaxis		
Yes	42	20.8
No	160	79.2
Acceptable compliance		
Yes	52	9.8
No	477	90.2
Total compliance		
Yes	21	3.9
No	508	96.1

**Table 2 antibiotics-12-00808-t002:** Characteristics of antibiotic administration to patients undergoing gynecological surgeries (*n* = 529).

Antibiotic Indication		
Yes	500	94.5
No	29	5.5
**Antibiotics used**		
Cefazoline	188	37.6
Ceftriaxone	145	29
Clindamycin	43	8.6
Clindamycin + Gentamicin	34	6.8
Ceftriaxone + Clindamycin	16	3.2
Ceftriaxone + Clindamycin + Gentamicin	13	2.6
Ceftriaxone + Gentamicin	6	1.2
Cefalexin	6	1.2
Cefazolin + Gentamicin	3	0.6
Cefazolin + Gentamicin + Chloramphenicol	3	0.6
Ciprofloxacin	3	0.6
Others	40	8
**Antibiotic regimen indicated**		
Single	415	83
Double	63	12.6
Triple	22	4.4
**Antibiotic prophylaxis**		
Yes	202	38.2
No	327	61.8
**Prophylactic antibiotics used**		
Cefazolin	116	57.4
Ceftriaxone	45	22.3
Chloramphenicol	13	6.4
Gentamicin	12	5.9
Clindamycin	10	4.9
Others	6	3.1
**Indicated prophylactic antibiotic regimen**		
Single	197	97.5
Double	4	1.9
Triple	1	0.5
**Antibiotic treatment at discharge**		
Yes	358	67.7
No	171	32.3
**Antibiotics indicated at discharge**		
Cefalexin	224	62.6
Dicloxacillin	21	5.9
Gentamicin	21	5.9
Gentamicin + Ceftriaxone	21	5.9
Ceftriaxone	16	4.5
Clindamycin	12	3.3
Others	43	11.9
**Indicated discharge antibiotic regimen**		
Single	318	88.8
Double	37	10.4
Triple	3	0.8

**Table 3 antibiotics-12-00808-t003:** Comparison of factors related to acceptable or total compliance in patients undergoing gynecologic surgery.

	Acceptable Compliance			Total Compliance		
	Si (%)	No (%)	*p*	Si (%)	No (%)	*p*
Age			0.12			0.56
Median (IQR)	30.5 (26–37.5)	33 (28–38)		35 (30–40)	33 (28–38)	
Hospital			<0.01			0.21
EsSalud	18 (5.8)	290 (94.2)		15 (4.9)	293 (95.1)	
Minsa	34 (15.4)	187 (84.6)		6 (2.7)	215 (97.3)	
**Comorbidities**			0.34			0.56
Si	52 (9.9)	469 (90.1)		21 (4.1)	500 (95.9)	
No	0 (0)	8 (100)		0 (0)	8 (100)	
Surgery			0.04			0.03
Cesarean	38 (12.6)	264 (87.4)		11 (3.6)	291 (96.4)	
Uterine curettage	11 (8.5)	118 (91.5)		10 (7.8)	119 (92.2)	
Hysterectomy	2 (3.6)	53 (96.4)		0 (0)	55 (100)	
Others	1 (2.3)	42 (97.7)		0 (0)	43 (100)	
Type of procedure			0.01			0.02
Emergency	40 (12.5)	279 (87.5)		18 (5.6)	301 (94.4)	
Programmed	12 (5.7)	198 (94.3)		3 (1.4)	207 (98.6)	
Anesthesia type			0.31			0.04
Regional	38 (10.6)	319 (89.4)		10 (2.8)	347 (97.2)	
General	14 (9.1)	140 (90.9)		11 (7.1)	143 (92.9)	
Sedoanalgesia	0 (0)	18 (100)		0 (0)	18 (100)	
Length of surgery (min)			0.32			0.08
Median (IQR)	80 (63.5–110)	90 (60–135)		75 (45–99)	90 (60–132.5)	

## Data Availability

The datasets generated and/or analyzed during the current study are available from the corresponding author upon reasonable request.

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
