# Peer review of "Compliance with Antibiotic Prophylaxis in Obstetric and Gynecological Surgeries in Two Peruvian Hospitals"

_antibiotics, 2023, doi:10.3390/antibiotics12050808_

Round 1

Reviewer 1 Report

Thanks for giving me the chance to review this manuscript. 

The study is interesting ,but the following issue should be discussed:

*The authors should clarify in the methodology section that the study is retrospective study.

*In sectio  of Limitations, the word "acknowledged "is not suitable and should be replaced with a better one.

*Why did the authors choose the duration of records during 2019? 

* Assesmebt of Compliance is complicated process, the authors should clarify in depth if high or complete compliance were related to the discharge from hospital?

Author Response

Dear Reviewer, thank you for your suggestions, please see our response in the attached file.

Reviewer 2 Report

Thanks for this study. Research on real-life abx use is highly relevant as this may direct stewardship interventions. I greatly enjoyed reading this manuscript, although it is striking to read that inadequate usage seemed to be that common. 

Could you clarify for me:

1) What was the prevalence of pre-eclampsia and HELLP syndrome in this cohort? I can hardly believe it is no more than the 4 patients with hypertension that you outlined in table 1. The prevalence of pre-eclampsia is generally 3-7%, although this may vary in different races, but I assume your cohort represents higher risk obstetrics given the need for C-sections.

2) Were any of the patients known colonizers with resistant pathogens (e.g., MRSA, CRO, etc.) that could explain alternative prophlyxis regimens (e.g., clinda + gent). What is the general prevalence of MRSA in your local population? 

3) Could you disclose/hypothesize the rationale for some prophylaxis regimens; e.g, ceftriax/clinda (did these patients have specific comorbidities), clinda/gent? 

4) As 14.8% of the cohort had other abx, I think it is prudent to specify the exact regimens. 

Author Response

(The authors gave the same response as above.)

Round 2

Reviewer 2 Report

Thanks for this revision.

A couple of comments from my side:

1. Line 102: what means "antimicrobial indications" in this sentence? 

2. Line 216-17: HELLP/pre-eclampsia itself is not an indication for abx beyond prophylaxis, it may just increase the risk on iatrogenic infections since these patients are often admitted for longer. Please resonate this in this section.

Author Response

Thanks for this revision.

Reply. Thanks for your invaluable suggestions.

A couple of comments from my side:

  1. Line 102: what means "antimicrobial indications" in this sentence? 
  2. We have improved the wording of this paragraph to indicate that 94.5% of patients received antibiotics.
  3. Line 216-17: HELLP/pre-eclampsia itself is not an indication for abx beyond prophylaxis, it may just increase the risk on iatrogenic infections since these patients are often admitted for longer. Please resonate this in this section.
  4. Thank you for your suggestion, we have added the above.